# On the development of sleep states in the first weeks of life

**Tomasz Wielek**[1,2]*, **Renata Del Giudice**[3], **Adelheid Lang**[1,2], **Malgorzata Wislowska**[1,2], **Peter Ott**[4], **Manuel Schabus**[1,2]*

**1** Laboratory for Sleep, Cognition and Consciousness Research, University of Salzburg, Salzburg, Austria, **2** Centre for Cognitive Neuroscience (CCNS), University of Salzburg, Salzburg, Austria, **3** Department of Health Sciences, Università degli Studi di Milano, Milan, Italy, **4** ITS Informationstechnik & System-Management, Salzburg University of Applied Sciences, Salzburg, Austria

\* wielekto@stud.sbg.ac.at (TW); manuel.schabus@sbg.ac.at (MS)

**Data Availability Statement:** Data are available at a public repository: https://figshare.com/articles/On_the_development_of_sleep_states_in_the_first_weeks_of_life_-_REVISED/9767567.

## Abstract

Human newborns spend up to 18 hours sleeping. The organization of their sleep differs immensely from adult sleep, and its quick maturation and fundamental changes correspond to the rapid cortical development at this age. Manual sleep classification is specifically challenging in this population given major body movements and frequent shifts between vigilance states; in addition various staging criteria co-exist. In the present study we utilized a machine learning approach and investigated how EEG complexity and sleep stages evolve during the very first weeks of life. We analyzed 42 full-term infants which were recorded twice (at week two and five after birth) with full polysomnography. For sleep classification EEG signal complexity was estimated using multi-scale permutation entropy and fed into a machine learning classifier. Interestingly the baby's brain signal complexity (and spectral power) revealed developmental changes in sleep in the first 5 weeks of life, and were restricted to NREM ("quiet") and REM ("active sleep") states with little to no changes in state wake. Data demonstrate that our classifier performs well over chance (i.e., >33% for 3-class classification) and reaches almost human scoring accuracy (60% at week-2, 73% at week-5). Altogether, these results demonstrate that characteristics of newborn sleep develop rapidly in the first weeks of life and can be efficiently identified by means of machine learning techniques.

## Introduction

Sleep of newborns greatly differs from the sleep of kids or adults. Adult-like classification of sleep into classical sleep stages is possible only from the age of 2–3 months onwards, since only then typical NREM patterns, like sleep spindles, K-complexes, or slow waves emerge (AASM; [1]). Until then the EEG landscape is dominated by low-voltage-irregular (REM/Wake), high-voltage slow (NREM/REM), mixed (Wake/NREM/REM) and tracé alternant patterns in NREM. Oscillatory activity of newborns is dominated also during wake by slow oscillations of a very high amplitude up to 100 µV [2]. Another hallmark of early brain activity is bursting

**Funding:** The study was supported by a grant from the Austrian Science Fund FWF (Y-777). TW, AL, MW, were additionally supported by the Doctoral College "Imaging the Mind" (FWF; W1233-G17). The funders had no role in study design, data collection and analysis, decision to publish, or preparation of the manuscript.

**Competing interests:** The authors have declared that no competing interests exist.

activity (known also as spontaneous "activity transients" or "delta brushes"), characterized by slow delta-like waves with superimposed fast, beta range activity [3, 4]. Hence in the first weeks of life the only discrimination of sleep stages possible is between wake, active sleep (REM) and quiet sleep (NREM) [5, 6].

Importantly the neonatal sleep-wake state organization also impacts on later development [7–9]. For instance sleep characteristics during the first postnatal days are related to cognitive development at an age of 6 months [10]. Being able to reliably characterize sleep in newborns has been recognized as crucial for both research and pediatric practice but is even to-date an inherently difficult endeavor.

Traditionally, sleep of infants is staged based on a visual inspection of polysomnographical (PSG) recordings, which often is supplemented by simultaneous observation of overt behavior and respiratory activity. However such manual sleep staging is time consuming, costly, requires high expertise, and can be quite variable due to varying staging criteria and/or noisy data. Crowell and colleagues [11] for example reported moderate inter-scorer reliability for infant sleep staging, with kappa coefficient going below 0.6, when staged according to the modified Anders manual [5]. Reliability can be improved by refining criteria for a specific age group individually, as done for example by Satomaa and colleagues [12], who reached kappa score of 0.73 indicating substantial agreement for one-month old babies. From a practical point of view however such fine tuning of sleep staging criteria might yield rather low reproducibility as the results cannot generalize to other age groups. We here try to address this issue and reached out for a more data-driven and objective analysis of EEG data using machine learning.

In the context of brain oscillations typically frequency-resolved information is exploited by means of spectral methods such as Fast Fourier Transform (FFT). More recently however, a stronger focus on irregular dynamics of brain signals gave rise to entropy-based features (for a review see [13]). Entropy quantifies the extent of irregularity in the EEG time signal, where repeating, predictable signal yield low entropy, while irregular, unpredictable signal yields high entropy. In contrast to the power spectrum capturing only linear properties of the brain signal, entropy-based features emphasize also additional characteristics of the EEG that are related to non-linear dynamics [14]. A non-linear behavior of human EEG was for example detected during adult sleep, especially in N2 stage [15, 16]. In contrast to FFT-based measures, symbolic measures such as permutation entropy are operating on the order of values rather than on the absolute values of a time series. This has a big practical advantage if a signal is highly non-stationary and corrupted by noise [17], as is the case with the data of newborns. For instance, noise due to high electrode impedance is less likely to affect symbolic measures such as permutation entropy [14].

Mounting evidence suggests that fluctuations in EEG entropy reflect both transient, state-like changes in human brain activity (e.g. wake and sleep states), as well as slower and more long lasting dynamics across a day or even brain maturation. Sleep studies investigating healthy adults report an overall trend of entropy decrease from WAKE, across transitional (N1) and light (N2) to deep (N3) sleep, with a relative increase during REM sleep [18–20]. A similar pattern was reported in newborns with higher entropy levels in active/REM sleep as compared to quiet /NREM sleep [21]. Diurnal changes in EEG entropy between daytime and night time periods–although diminished in size in relation to healthy individuals–were also found in patients following a severe brain-injury [22].

Across the development from childhood to adulthood a permanent increase of EEG entropy has been observed [23]. Also in adults (19–74 years) an age-related increase of EEG entropy was reported [24]. Interestingly, during the first weeks of infancy, this pattern is much less consistent, and may even be accompanied by transient declines in EEG entropy. For

instance, Zhang and colleagues [25] reported EEG entropy during sleep to increase across the first month of baby's life. Then suddenly this patterns changes, such that entropy remains constant during quiet sleep, but decreases during active sleep. This transient change in entropy evolution is in good agreement with previous results showing a general decrease in high frequency (beta band) power occurring within the first month of life ([26, 27]also for sleep stage specific findings see [28]).

To delve deeper into the dynamically changing landscape of early brain activity, we analyzed newborn sleep EEG data (PSG and hdEEG) of 42 participants at week 2 and 5 after birth. It has to be mentioned that the reported data were not recorded during continuous night-time sleep period, thus may differ from natural sleep in newborns. First we evaluated sleep that was previously scored visually, in terms of both entropy and oscillatory power. In contrast to Zhang and colleagues, where temporally unspecific entropy was used, we quantify entropy over multiple temporal scales, with the aim to add knowledge, especially in light of classic, 'frequency-resolved' findings. Second we aimed at testing the possibility of automatizing sleep staging by using the previously extracted entropy measures in a machine learning approach. This approach could ultimately complement or even replace visual staging and thus make the sleep scoring in newborns more objective and replicable. Last but not least, we tried cross-session classification such that we could assess whether our algorithm can generalize between age groups and reveal the similarity or "dissimilarity" in sleep organization this early in life.

## Participants and methods

### Participants and EEG recording

Mothers of 42 full-term infants (15 female) were recruited for a study on prenatal learning. Polysomnography (PSG) was recorded from all but one newborn during two separate sessions: first at 2 weeks (14.8±4.3 days) after birth and then 5 weeks (36.7±4.3 days) after birth. Recording took place in the home environment of the mother and infant. EEG were recorded with an ambulatory, high-density (128-EEG channels cap) system using a Geodesic Sensor Net (Geodesic EEG System 400, Electrical Geodesics, Inc, Eugene, OR; US). The signal was recorded continuously with a sampling rate of 500Hz over 35min (n = 11) or 27min (n = 31). Recording times were determined by the experimental protocol including nine 3min or 5min periods of alternating rest and auditory stimulation periods (with simple pre-recorded nursery rhymes). For the current study we disregard this experimental stimulation and merely focus on the changes in behavioral states over the full recording time. The study was approved by the ethic committee of the University of Salzburg (EK-GZ 12/2013) and all parents provided written informed consent before participation.

### Data preprocessing

Preprocessing was done in Brain Vision Analyzer (Brain Products GmbH, Gilching, Germany, version 2.0) and MNE-python software (version 0.16.1). Data was down-sampled to 125 Hz and re-referenced to average reference of all the 128 EEG channels. EEG was band-pass filtered (FIR filter with hamming window) between 0.5 and 35 Hz. Electrode (impedance check) artifacts characterized by a 20Hz component were deleted semi-automatically by first visually inspecting individual recordings in the time-frequency domain and next iterating over segments. 95% percentile thresholding was used to exclude bad segments which resulted in an exclusion of 4.5% of total segments. Note that this step was performed in addition to the exclusion of segments staged as movement or transitional sleep by the human scorer. The high density EEG montage was then subsampled to the equivalent of a habitual sleep montage with only 6 EEG channels (F3, F4, C3, C4, O1, O2) and 5 peripheral channels: bipolar ECG, bipolar

EMG, bipolar VEOG, as well as a HEOG left, and HEOG right both re-referenced to the right ear as recommended in [29]. For the subsequent automatized sleep scoring we used the same reduced PSG setup as we used for the visual sleep scoring. This allows making a fair comparison between the two sleep scoring approached on one hand, and increases applicability of our classifier to other baby PSG datasets on the other hand.

## Visual sleep staging

Eighty-four recordings were visually sleep-scored by an expert sleep scorer (Scholle) according to scoring criteria [30] based on 30-second PSG segments. Each segment was assigned to one of the five classes: quiet sleep (NREM), active sleep (REM), wake (WAKE), movement time, and transitional sleep. To account for possible difference in the amount of movement from week-2 to week-5 all epochs scored as movement (as well as transitional sleep) were excluded from further analysis (7.7% of segments). Ten recordings were considered "unscorable" by our expert and removed from further analysis. In the next step we validated the manual scorings by examining the simultaneously recorded videos. We followed established Prechtl staging criteria [31]. Whenever we detected mismatch between PSG-based scorings and video recordings, for example observed infant's open eyes in epochs staged as sleep, we sought for a consensus score. Due to technical issues (EEG signal corrupted) we excluded one of the recording sessions of one participant. All in all 72 recordings (34 at 2 weeks of age; 38 at 5 weeks of age) were included in the final analyses.

## Entropy measure

Entropy quantifies the irregularity or complexity of signal fluctuations, where repeating, highly predictable signal yield low entropy, while irregularity yields high entropy. We used permutation entropy (PE) as a robust entropy measure that first converts EEG time series into a sequence of data-patterns (where each pattern describes the order relations between neighboring EEG voltages), and next quantifies the distribution of these patterns by using the Shannon entropy equation (cf. Fig 1a). Highest PE (maximal information) is attained when all patterns have equal probability. Further generalization of this method, called multi-scale permutation entropy (MSPE), [32] applies coarse-graining to the original broadband signal by averaging data within non-overlapping windows [33]. Like low-pass filtering, coarse-graining eliminates fast fluctuations from the signal biasing the complexity estimates towards increasingly slower time scale (cf. Fig 1b).

MSPE was calculated for non-overlapping 30s segments, for each PSG channel separately, and for 5 different levels of coarse-graining (scales). To maximize the predictive power of the classifier we used all 5 temporal scales as an input. This resulted in 55-dimensional feature vectors (per segment) for each subject. The classifier is later called as MSPE-based. For univariate analysis reported later we used MSPE computed at a scale of 1 (original signal, mixture of fast and slow temporal scales) and at a scale of 5 (fast temporal scales eliminated), later referred to as fast scale and slow time scales respectively. Analysis for intermediate scales was included in supplementary materials for completeness (cf. S6 Fig).

## Spectral measure

We also calculated power spectral density (PSD) for the same 30s segments in 1 Hz steps and for frequencies between 1–30Hz. Welch's method with overlapping Hamming windows was used. Similarly to MSPE analysis, we used all frequency bins as an input for the classifier (later called PSD-based) providing 330-dimensional (11 x 30) feature vectors (per segment) for each subject.

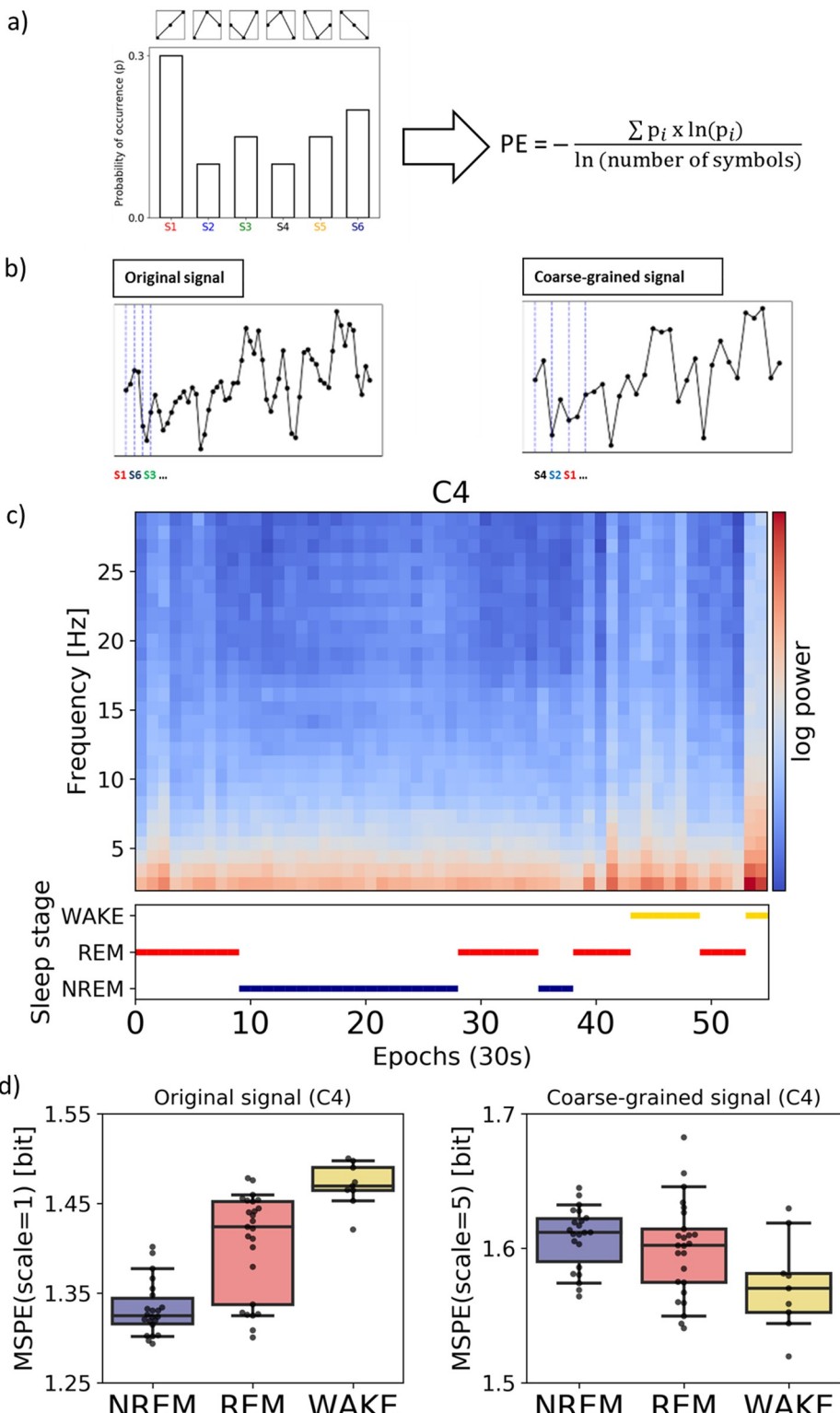

**Fig 1. Multi-scale permutation entropy as useful feature for neonatal sleep stage classification.** (a) Six possible ordinal patterns are identified (p*i*), their distribution is formed and Shannon entropy is computed. (b) Simulated time series prior (left) and after the coarse-graining procedure (right) are shown. Note that the granulation removes the fast-varying changes, which allows estimating the entropy on a slower temporal scale. (c) PSDs of individual epochs for a single recording (week-5) and corresponding visual/manual sleep staging, log-transformed power is shown for better

visibility. (d) Corresponding MSPE values for the same recording depicted separately for the original signal (left) and after coarse-graining (right). Note that these three classes or stages are distinguishable on both scales, yet different patterns are observed in the original and coarse-grained computation.

## Statistical analysis

Univariate statistical analysis was performed for both entropy measures as well as PSD estimates. To compare entropy data linear mixed models were used as they are more suited to deal with unbalanced datasets than repeated measure ANOVA. This is of a particular benefit as the number of available segments was limited (for example NREM for week 2; cf. Table 1). To provide an equal number of observations for each subject and for each sleep stage a bootstrap sample of 10 MSPE values was repeatedly (1000 times) drawn and then averaged. Matrix of MSPE values (sleep stage x participant x session x location) entered the model as a dependent variable. Two sessions (week-2 and week-5), three sleep stages (NREM, REM, WAKE) and three locations (frontal, central, occipital) served as fixed effects, and participants as a random effect. We also included a random slope to account for inter-individual differences in complexity of each baby from week-2 to week-5. To select the model with optimal fit the Akaike information criterion was applied. All model parameters were estimated using restricted maximum likelihood estimation. Wald Chi-Squared test was used to test for significance of the model variables. Two independent testings were performed with MSPE at scale of 1 and at scale of 5 as a dependent variable. Additionally, we carefully report differences if the exclusion of statistical outliers (as identified by the interquartile range rule) changed results significantly. Linear mixed model analysis was performed with the lme4 package [34] by using the statistical software R 3.4.0 [35]. Post hoc, multiple comparison procedure was performed with Tukey test using the glht-method of the multcomp package [36].

To compare power spectra between the two sessions we first used similar bootstrapping as for MSPE and then ran cluster-based permutation tests as implemented in MNE-python software (version 0.16.1).

## Machine learning

The principle behind supervised machine learning (ML) is to train a predictive model, by automatically encapsulating information from previously labeled dataset (in our case visual sleep staged PSG epochs). We performed an epoch-by-epoch classification into one of three distinct

**Table 1. Sample descriptives.**

| Session | Age (in days) | | Sleep stages (visually scored 30s epochs) | | Subjects | MSPE Fast | MSPE fast | MSPE Slow | MSPE slow |
|---|---|---|---|---|---|---|---|---|---|
| | Mean | SD | Name | n | % | Mean | SD | Mean | SD |
| **week-2** | 13.9 | 3.7 | NREM | 213 | 35 | 1.356 | 0.05 | 1.609 | 0.031 |
| | | | REM | 1004 | 97 | 1.435 | 0.048 | 1.615 | 0.032 |
| | | | WAKE | 431 | 62 | 1.451 | 0.053 | 1.585 | 0.045 |
| **week-5** | 36.2 | 3.9 | NREM | 387 | 66 | 1.315 | 0.044 | 1.58 | 0.051 |
| | | | REM | 1056 | 95 | 1.392 | 0.042 | 1.606 | 0.028 |
| | | | WAKE | 392 | 66 | 1.44 | 0.056 | 1.569 | 0.036 |

The analyzed data set consisted of two recordings from a group of healthy newborns with three sleep stage classes that were identified using visual scoring. EEG signal complexity was estimated using multi-scale permutation entropy (MSPE). The subjects% column reflects the percentage of participants who actually showed a specific sleep stage in the respective (week-2 or week-5) recording sessions. MSPE values refer to the results from frontal channels (F3 and F4).

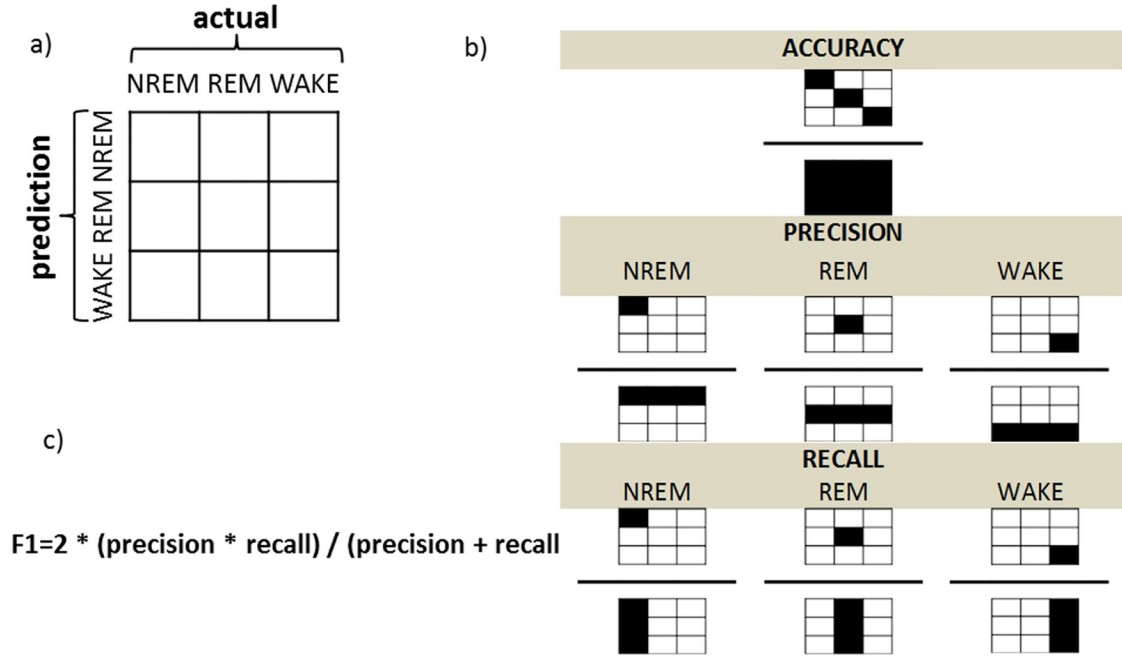

**Fig 2. Evaluation of the performance.** (a) The confusion matrix relates the actual (visual) scoring to the classifier predictions. (b) Values from the confusion matrix were used to calculate the overall accuracy and the class-specific F1 scores. Note that diagonal corresponds to agreement between the visual sleep staging and the MSPE-based automatic classification. Whereas accuracy (top panel) reflects relative amount of agreement between predicted and actual scores, F1 accounts for precision (middle panel) and recall (bottom panel). Precision (known also as positive predictive value) takes into account false positives and is defined as the ratio of epochs classified by both the classifier and the human scorer as given sleep stage to all of the epochs that the classifier assigned as that sleep stage (b, middle panel). Recall (known also as sensitivity) in turn takes into account the amount of false negatives and is defined as the ratio of epochs classified by both the classifier and the human scorer as given sleep stage to all of the epochs that the human scorer classified as the given sleep stage (b, lower panel). (c) The F1 score combines the two measures into a single metric.

sleep classes: NREM, REM, WAKE. A random forest (RF) classifier was used as a main component of the classification pipeline. We employed repeated (10 repetitions) two-fold cross-validation by indexing half of the subjects as testing and the other half as training subjects. It has been shown that this procedure has smaller variance than the typical leave-one-subject-out cross validation [37]. Training and testing sets were created by concatenating data of all subjects assigned to one of the two groups. Random under sampling of epochs was performed to equalize the number of epochs across sleep class, both in the training and testing set. Additional cross-validation with the training set was performed to find optimal configuration of the classifier parameters (hyper-parameters). Finally we used two types of the two-fold cross-validation: (1) within sessions such that both training and testing subjects were of the same age, (2) across sessions such that training subjects and testing subjects were of different age (cf. S2 Fig). Importantly both procedures were run as a between-subject classification, such that each participant was assigned to either the training or testing set (i.e., no two sessions of the same subject in training and testing at the same time). Each cross-validation was repeated 20 times, and the median scores (accuracy, F1-score) are reported. Chance level of the accuracy scores was estimated by running each cross-validation on shuffled data (100 repetitions). The machine learning analysis was performed in Python using the scikit-learn package [38]. To evaluate the performance of a classifier standard performance metrics were used. Class-wise performance (for NREM, REM and WAKE classes separately) as well as the overall performance was accessed using the F1 score and the accuracy score respectively (cf. Fig 2).

To compare output of the classifiers based on different data both a MSPE- and PSD-based classification was repeated 20 times on the full dataset (i.e., the week 2 and week 5 data merged). The corresponding accuracy scores of MSPE- and PSD-based classifiers were compared statistically with Mann-Whitney U test. Since the MSPE-based classifier had significantly better performance as compared to the PSD-based classifier (cf. S1 Fig), we restricted our subsequent classification analysis to MSPE classifier results.

## Results

To assess sleep rhythm of newborns EEG recordings in a sample of 42 participants were used. Final analysis included 72 visually sleep scored PSG recordings divided into two age groups: 2-weeks-old (N = 34) and 5-weeks-old (N = 38). We evaluated changes in sleep stage distribution (including NREM, REM and WAKE stages) from the age of 2 weeks to the age of 5 weeks. On average 5-week old newborns spend a higher percentage of total time (19%) in NREM sleep as compared with 2-weeks old babies (11.5%). In contrast relative REM duration decreases from 60.6% at week-2 to 57.2%, at week-5, and WAKE decreases from 27.9% at week-2 to 23.8 at week-5. A significantly larger proportion of participants showed NREM during week-5 (66%) as compared to week-2 (35%) ($\chi^2$ (1) = 6.81, p < .05). Using paired-samples Wilcoxon test (by including only those subjects that actually show given sleep-state in both sessions), we found no significant differences in the median duration of classes from week-2 to week-5 (Wilcoxon Signed-Ranks tests; NREM: Z = 10.5, p = .15, REM: Z = 191.0, p = .56, WAKE: Z = 32.5, p = .21). Subsequently, EEG signal complexity was investigated across different sleep stages and recording sessions. Please see Table 1 for a detailed overview with respect to age, distribution of sleep stages and the complexity measure–multi-scale permutation entropy.

### Entropy and spectral measures

We observed that the multi-scale permutation entropy (MSPE) at a fast scale (i.e., no coarse-graining, incl. mixture of low and faster frequencies) as well as slow scale (i.e., fast temporal scales eliminated) significantly differed between sleep stages, recording sessions as well as channel locations. Table 1 and Fig 3 illustrate the results for MSPE averaged over F3 and F4 channels (for central and occipital sites see S4 Fig).

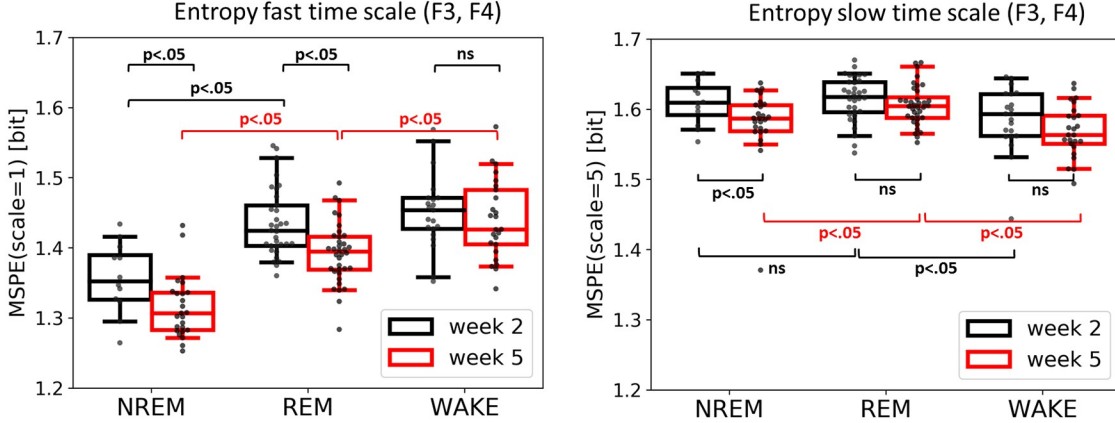

**Fig 3. Entropy at both fast (left) and slow (right) time scale across sleep/wake states and the two recording sessions. MSPE values were averaged over frontal electrodes**. Note that at the fast scale all three stages are distinguishable at week 5, but not yet at week 2. At both scales there is a clear difference in signal complexity between week 2 and week 5 with week 2 being generally higher in entropy.

At a fast time scale (Fig 3, left) there was a main effect of session (week-2 vs week-5: Wald chi-square (1, 73) = 27.65, p < .001) indicating higher permutation entropy during week-2 as compared to week-5. Furthermore we observed a main effect of sleep stage (NREM vs REM vs WAKE: Wald chi-square (2, 73) = 237.64, p < .001), and a main effect of channel location (frontal vs central vs occipital: Wald chi-square (2, 73) = 45.71, p < .001). There were significant interactions between session and sleep stage (Wald chi-square (2, 73) = 7.79, p = .02) as well sleep stage and location. A post-hoc Tukey test for the factor sleep state confirmed higher entropy during WAKE compared to both REM (mean(SE) = .021(.005); Z = 4.09, p < .001) as well as NREM (mean(SE) = .086(.006); Z = 13.93, p < .001). In addition there was also significant difference between REM and NREM (mean(SE) = .065(.005); Z = 12.03, p < .001). A post-hoc Tukey test for the factor location yielded lower entropy over frontal channels as compared to the central location (mean(SE) = -.024(.006); Z = 4.18, p < .001), as well as lower entropy over frontal as compared to occipital sites(mean(SE) = -.029(.006); Z = 5.0, p < .001). A post-hoc Tukey test on the interaction between session and sleep stage revealed higher entropy during WAKE compared to REM, but only at week-5 (mean(SE) = .031(.007); Z = 4, p < .001). A post-hoc Tukey test on the interaction between sleep stage and location yielded higher entropy during WAKE compared to REM, only over the frontal location (mean(SE) = .032(.008); Z = 4.18, p < .001). Note that the session x sleep stage interaction changed to a trend after outliers (based on the interquartile rule) were excluded (Wald chi-square (2, 73) = 4.67, p = .09).

At a coarse time scale (Fig 3, right) we found a similar main effect of session (week-2 vs week-5: Wald chi-square (1, 73) = 6.96, p = .008) indicating an overall decrease in EEG complexity from week-2 to week-5. Additionally we observed a significant main effect of the sleep stage (NREM, REM, WAKE) (Wald chi-square (2, 73) = 87.4, p < .001). A post-hoc Tukey test for the factor sleep state revealed higher entropy during REM compared to NREM (mean(SE) = .02(.004); Z = 5.47, p < .001), also higher entropy during REM compared to WAKE (mean(SE) = .03(.004); Z = 8.81, p < .001), as well as higher entropy during NREM compared to WAKE (mean(SE) = .01(.004); Z = 2.42, p = .04) Moreover, we observed that the pattern of relative entropy levels (across sleep stages) reverses at coarse temporal scale (Fig 3, right) compared to fine temporal scale (Fig 3, left); such that WAKE shows the lowest entropy level for coarse temporal scale. The interaction between session and sleep stage was found to be marginally significant (Wald chi-square (2, 73) = 6.0, p = .051). A post-hoc Tukey test showed that NREM is distinguishable from REM but only during week-5 (mean(SE) = .03 (.005); Z = 5.79, p < .001).

Similarly, we also evaluated changes in power spectral densities (PSD) across sleep stages and recording session. We found an increase in spectral power from week-2 to week-5, but only during NREM (Fig 4, left). This pattern was observed also for central electrodes and to a limited degree (2–4Hz frequency bins) for occipital ones (cf. S5 Fig). For Spearman's rho correlations between spectral and entropy features please see S9 Fig.

## Machine learning classification

**Influence of the feature extraction on the classification performance: PSD vs MSPE.** Multiscale permutation entropy improves the overall classification accuracy as compared to PSD (cf. S1 Fig). The per-class evaluation shows that MSPE improves performance in the discrimination of all classes, but in particular for WAKE (73% instead of 59% median accuracy) wherefore we decided to focus on MSPE in all subsequent classification analyses.

**Separate classification for week-2 and week-5—Within age-group.** To automate sleep staging we used machine learning and previously extracted entropy features. We first

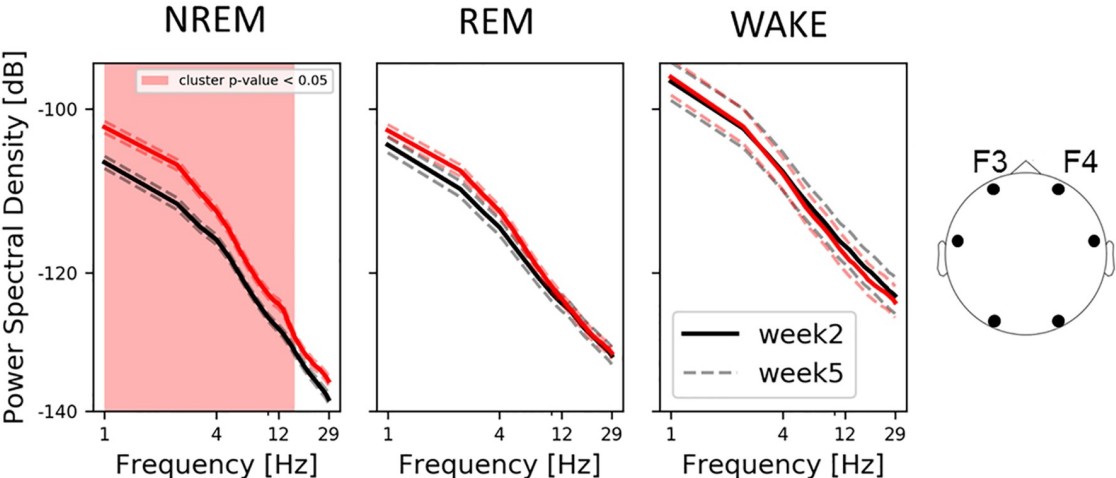

**Fig 4. Average log-log-scale PSD spectra per sleep stage over frontal electrodes.** The shaded area highlights statistical differences between week-2 and week-5 recordings. The dashed lines represent the standard error of the mean. Note that only NREM shows differences in the PSD spectra between age groups and the developing 9–14 Hz peak exclusively observed at week-5 of age.

performed within age-group classification and evaluated results from the two independent and age-specific classifiers. That is, we computed the performance scores of a classifier that was both train and tested on data from either (1) weekd-2 or (2) week-5 recording. Accuracy scores across all 3 sleep stages were significantly higher than would be expected by chance. In randomization test across 3 classes, chance level was at ~33%. For both the week-2 and week-5 classification (cf. Fig 5, left upper panel). Moreover, the classifier performed better on the week-5 (Mdn = 72.7%) as compared to the week-2 (Mdn = 60.1%) babies (U = 1, Z = 6.5,

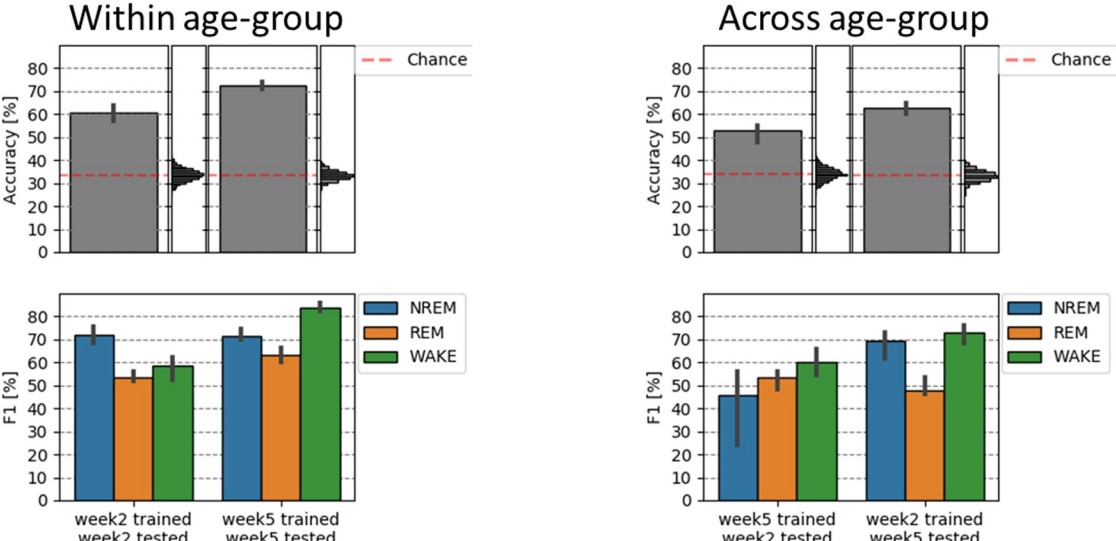

**Fig 5. Classification of sleep states in the week-2 and week-5 old newborn (left panel) and its generalization across age (right panel).** Sleep in older (5 week old) babies can be classified more accurately as compared to 2 weeks (left panel). Right panel shows that the "generalization" or classification across age-groups leads to lower classification accuracy specifically for detecting stage NREM (week-5 training, week-2 test), but also stage REM (week-2 training, week-5 test). Vertical histograms represent the null distribution, with the empirically estimated chance levels in both cases being close to 33% (red dashed line). 95% confidence-intervals (for both accuracy and F1 scores) are displayed on the basis of bootstrap analysis.

p<0.001). The per-class evaluation (cf. Fig 5, left lower panel) shows higher F1 classification scores during week-5 (Mdn = 63.1%) as compared to week-2 for REM (Mdn = 53.2%); (U = 24, Z = 5.2, p<0.001), as well as for WAKE (Mdn = 83.6% vs. Mdn = 58.5%; U = 1, Z = 6.6, p<0.001).

A confusions matrix (cf. S3 Fig) as well as visual inspection of the single subject results (cf. S7 Fig for an exemplary subject) reveals that a very limited proportion of NREM epochs is actually falsely classified as WAKE (on average 3%) or vice versa (8% of the epochs). During week-2 the classifier is worst in distinguishing REM and WAKE, while at week-5 the classification generally increases and NREM and REM are sometimes wrongly assigned.

**Cross classification between week-2 and week-5—Across age-groups.** In a second and last step we tested whether the classifier can generalize across age groups. That is we trained the classifier on a given age-group and looked at the classification accuracy onto the other age-group (cf. Fig 5, right).

Compared to the within-session classification (week-2 to week-2 and week-5 to week-5), the cross-session (week-5 to week-2 and week-2 to week-5) classification accuracy was decreased by 7.7% and 9.9%, respectively. Classification however remained well above chance level (~33%) and was better when the classifier was trained on week-2 and tested on week-5 data (Mdn = 62.8%) as vice versa (Mdn = 53.1%) (cf. Fig 5, right). Interestingly, the per-class evaluation shows that detection of NREM deteriorates when trained on week-5 (and tested on week-2) data, and REM classification deteriorates when trained on week-2 (and tested on week-5) data only. The classification of state WAKE remains relatively stable indicating that the way this stage presents itself in the EEG remains widely constant from week-2 to week-5 in the newborn.

**Uncovering the black box of classification– feature importance and decision boundaries.** Evaluation of channel importance from a trained random forest classifier revealed that primarily the horizontal EOGs, frontal brain channels (F4, F3) as well ECG contribute to a good classification accuracy (S8 Fig). Visualization of classification boundaries of a trained random forest classifier in addition confirms previous results and reveals more compact and distinct sleep/wake classes at week-5 as compared to week-2 (see S8 Fig for details).

## Discussion

The main focus of this study was the development of an automatic sleep staging technique in order to sleep stage newborn sleep as early as 2 to 5 weeks after birth. As babies at this age also sleep a significant proportion of time (and irrespective of environmental stimulation such as noise) we also had the opportunity to study sleep and associated brain dynamics in this early age group. However, please note that we do not claim that our newborn sleep data is necessarily representing natural sleep at that early age as auditory stimulation is ongoing half of the time in our study protocol.

Generally newborns are known to sleep up to 16–17 hours a day. We found that the most dominant behavioral state in our study is actually active sleep or state REM (week-2: 60.6%, week-5: 57.2% of total time). Indeed, several earlier studies have shown that newborns spend more than half their time in REM [2]. Its proportion, however, gradually decreases within the first 12 weeks of life [39], which is interpreted as an ongoing adaptation of the sensory system to the environment [2]. In contrast the mean percentage of time for NREM increases from week-2 (11.5%) to week-5 (19%). This proportional increase in NREM is believed to reflect a gradual shift towards the adult pattern known as slow wave sleep (SWS) [2] which is known to be important for recovery as well as brain plasticity or learning. However, it is worth mentioning that testing longitudinally within subject, we found no significant changes in the

percentage of NREM. It was likely due to limited sensitivity in this measure, magnified by small number of available subjects with NREM during week 2.

Over the first weeks of life the human brain is growing at a rapid rate establishing a complex network that includes trillions of synaptic connections [40]. Accordingly, a continuous increase in brain signal complexity could be expected. Instead, we observed a clear decrease in EEG complexity from week-2 to week-5 in present data. Although similar findings have already been reported for both entropy [25] as well as spectral [26–28, 41] EEG brain measures, the understanding of what might account for this effect remains limited. Most of the aforementioned authors point to a decline in bursting activities (known also as spontaneous "activity transients" or "delta brushes"), which are abundant in premature infants, but which remain detectable until about the end of the first month of life [42].

In contrast to prior approaches, where temporally more unspecific entropy measures were used [25], we used multi-scale entropy (MSE) providing more details about temporal scale or frequency band which may contribute to the effect. In our data we found that EEG entropy decreases with age (especially during NREM and REM). This change was observed not only at the fast temporal scale, (mixture of low and high frequencies), but also at the slow temporal scale (slow frequencies only), suggesting a specific bandwidth or temporal scale being involved. Indeed, we also observed a significant increase in EEG spectral power at 1–15Hz (during NREM). The observed entropy decrease and corresponding power increase is likely linked to the emergence of sleep spindles taking place between week-2 and week-8 after birth [41]. The fact that frontal channels have been identified as the ones with lowest entropy (fast temporal scale) could be related to infant (1.5–6 months) sleep spindles which are prominent over the fronto-central area [43]. Nevertheless, this result needs to be interpreted with caution. It is widely acknowledged that frontal EEG channels are impacted most by eye blinks and muscles artifacts. Despite using robust entropy measure, we cannot exclude the possibility that some part of the effect may be driven by non-neural sources.

Most of the before mentioned studies focus exclusively on sleep periods (as it constitutes about 70% of the time in newborns from birth to 2 months), and leave out periods of wakefulness. It was therefore rather unclear whether foremost "quiet" NREM and "active" REM sleep states show these extensive changes early in development or if similar changes in the "brain signatures" are found in wakefulness during the first weeks of life. Indeed, data indicate that a drop in entropy at the fast temporal scale is exclusive to NREM and REM sleep states, with no significant changes during wakefulness. Likewise, we observed an increase in EEG spectral power only during NREM. These findings support the first line of interpretation, as in adults, newborns sleep spindles are observed during NREM sleep [2, 41]. Interestingly, the age-related decrease in entropy at a slow temporal scale is also observed only for NREM which supports the idea of a reorganization of brain oscillations, and the formation of more stable sleep-wake cycles with dominant slow waves during NREM.

Finally, in adults the relative entropy level across sleep stages is strongly time-scale dependent. At fast temporal scale EEG entropy follows a pattern of WAKE > REM > N2 > N3 whereas at slow temporal scale this pattern reverses (N3 > N2 > N1 > REM > WAKE) and is interpreted as a reflection of mainly local information processing during WAKE and increasingly global synchronization during NREM (reaching its culmination during N3) [15]. Our result shows that in neonates this pattern reverses partially. Both NREM and REM sleep show higher entropy than WAKE, however there is no significant difference between NREM and REM at the slow temporal scale which suggests that global connections may not be yet fully functional likely due to incomplete myelination of the newborns brain [44]. In relation to PSD data, we observed that high frequency spectral components correspond to higher entropy

(when compared across sleep stages), which agrees with previous studies suggesting a link between fast scale entropy and PSD slope [19, 24].

In addition to this descriptive analysis of baby EEG data in the first weeks of life, we developed a neonatal sleep classifier by employing machine learning using entropy-based features. We reveal that our classifier performs well over chance (60.1% week-2, 72.7% week-5) and is close to human scoring performance with adapted scoring rules (inter-scorer agreement of 80.6%, kappa score of 0.73 [12]). Crowell and colleagues [11] for example reported moderate inter-scorer reliability for infant sleep staging, with kappa coefficient going below 0.6.

However it has to be noted, that visual sleep scoring of neonatal recordings is particularly difficult, even for experienced sleep experts. In order to reach an acceptable inter-scorer agreement intensive training and careful attention to scoring specifications (which even vary in the literature) are required [11], which undermines the practical applicability of this approach. Note that even in adults the interrater agreement rarely exceeds 80% agreement [45]. As a matter of fact, 12 out 72 recordings in our sample have been classified as "difficult" by our scoring expert (Scholle S.;[30]). Considering the uncertainty in the 'ground truth', the obtained results are notably high. In fact after excluding the 12 "difficult" recordings the overall classification raises by about 6% which confirms presence of mislabeling in the dataset. The high classification accuracy of our automatic classifier also demonstrates that complexity measures are able to capture and quantify essential characteristics of vigilance states in newborns. Recently an independent group of researchers has used a similar approach to study sleep in preterm infants finding a significant correlation between the EEG complexity and the infant's age (ranging from 27 to 42 weeks) [46]. In contrast to the aforementioned study (where sample entropy was used), our approach was based on robust permutation entropy. A limitation of our current approach is certainly that we did not yet include respiration (mainly due to practical reasons, e.g., to ensure a rapid recording-start in the newborns) and that our "ground truth" scoring is lacking a double scoring. Given that we included a pioneer (Scholle S.) in infant sleep staging which carefully reviewed all our baby PSG recordings (including simultaneous video recordings and Prechtl vigilance-scorings from the recordings) we believe that our manual scoring is as good as it can get for the moment. Future studies should still try to add respiration and independent second scorers which potentially would lead to further improvement of classification accuracy.

The significantly higher overall classification performance for week-5 as compared to week-2 recordings indicates that the neonatal sleep/wake states become more distinguishable within a very short time spanning only about 3 weeks of early human development. For across age-groups classification we observed overall decrease in accuracy as compared to within-group classification, which indicates marked "dissimilarity" in the sleep organization between the two sessions. It is worthy of note that although fully consistent with our univariate entropy results, the classification approach models simultaneously all temporal scales and all channels (including EMG, ECG, EOGs), which provides a more complete picture on the entropy changes across time. Hence, to account for the rapid developmental changes in newborns and thus further improve the automated sleep classification, we suggest using 'transfer learning' procedure, known from recent application in deep-learning. In this scenario a classifier could be pre-trained on larger dataset and further fine-tuned on a specific age-group.

Interestingly testing on week-5 (when trained on week-2) is outperforming testing on week-2 (when trained on week-5) classification. This seems surprising as in case of cross-session (or -condition) classification it is in general more efficient to train a classifier on data with clear class-boundaries (high signal-to-noise; in our case week-5) and test it on data with less definite class-boundaries (low signal-to-noise; in our case week-2) than the other way around

[47]. In our case however, this difference in signal-to-noise ratio has been counterbalanced by the sleep specific entropy decrease from week-2 to week-5, which manifests itself as decreased performance in classifying both NREM (week-5 training, week-2 test) and REM (week-2 training, week5-test) with a 24% and 14% accuracy drop, respectively. Interestingly there is no change for stage wake when applying cross-session classification. This suggests that developmental changes in signal complexity are more pronounced for NREM and REM states whereas stage wake may be already widely developed.

Finally yet importantly we found that horizontal EOGs, frontal brain channels as well ECG contribute the most in the classification. This suggests that development of sleep patterns is not only associated with neural, but also with physiological changes (e.g., eye movement) and that these features may be crucial markers for the manually labeled sleep stages. In summarizing we observed massive developmental changes on the brain-level in the first 5 weeks of life in human newborns. These changes were limited to "quiet" NREM and "active" REM sleep and showed an unexpected drop of signal complexity from week-2 to week-5. In addition our classifier data demonstrated that we can classify well above chance and similar to human scorers using multi-scale permutation entropy (using just 6 EEG and 5 physiological channels). Altogether, these results highlight the need to perform electrophysiological studies during the first weeks of life where rapid changes in neuronal development and related brain activity can be observed.

## Supporting information

**S1 Fig. Influence of the feature extraction procedure on the classification performance.** Multiscale permutation entropy as compared to PSD boosts discrimination of sleep stages. MSPE improves classification especially of WAKE, also REM and slightly NREM class (note the diagonals of the lower panels).
(TIF)

**S2 Fig. Cross-validation schemes.** Splitting into training (in blue) and testing (in green) sets was performed within sessions (upper row) or across sessions (lower row). Note, that half of the subjects are used to train and half to test (two-fold cross validation), with both sessions of a single subject (week 2 and 5) always being in separate sets.
(TIF)

**S3 Fig. Confusion matrices summarizing the classification results for week-2 and week-5 (MSPE-based classifier).** Note the off-diagonals showing limited proportion of NREM falsely classified as WAKE (on average 3%) and similarly WAKE falsely classified as NREM (on average 8%).
(TIF)

**S4 Fig. Comparison of MSPE at fast time scale between sleep stages and the two recording sessions (week-2 vs week-5 data)—Central and occipital channels.** Note that overall entropy at the fast temporal scale is lower over frontal (see Fig 3, main text) as compared to both central and occipital channels (left panels).
(TIF)

**S5 Fig. Average log-log-scale PSD spectra for the individual sleep stages for central and occipital electrodes.** The shaded area shows statistical difference between week-2 and week-5. Note that similarly to frontal channels (main text), there is a clear difference in PSD also for central channels during NREM.
(TIF)

**S6 Fig. Comparison of multi-scale permutation entropy values across different scales between sleep stages and sessions.** Points represent the averages and error bars show the 95% bootstrap confidence intervals. Note that results for scale = 1 and scale = 5 (X-axis) correspond to the results presented in the main text. Maximal entropy in NREM and WAKE (also REM) is attained at different temporal scale (X-axis). Also, similarly to the main text there is no statistical difference in WAKE between sessions (week-2 vs week-5) across all temporal scale.
(TIF)

**S7 Fig. Exemplary single-subject classification (week-5).** Note the limited proportion of NREM epochs being falsely classified as WAKE and similarly WAKE classified as NREM.
(TIF)

**S8 Fig. Channel importance extracted from a trained random forest classifier and a decision boundaries of a trained random forest classifier.** Horizontal EOGs, frontal channels as well ECG contribute most to the sleep classification (upper panel). For visualization purposes multidimensional scaling was used to reduce dimension of the MSPE data to two (lower panel, X and Y axis). Points represent epochs (N = 100 for each class), colors (red, green and blue) represent true class labels and shading (pink, light blue and light grey) shows the decision boundary. Note that in week-5 (right panel) there is more apparent overlap between the true class labels (points) and the predictions (shading) as compared to week-2, which agrees with higher classification accuracies for week-5.
(TIF)

**S9 Fig. Correlations between spectral features and entropy at fast temporal scale (both sessions were merged).** Spectral features correspond to the average power values within three frequency ranges (rows). Solid red line indicates significant results. Note negative correlation between entropy and delta-theta band power during NREM.
(TIF)

## Acknowledgments

We are especially grateful to S. Scholle for manual sleep staging and A. Lang for manual verification of all sleep scorings, with Prechtl scores and simultaneous video files.

## Author Contributions

**Conceptualization:** Tomasz Wielek, Manuel Schabus.

**Data curation:** Tomasz Wielek, Malgorzata Wislowska, Peter Ott.

**Formal analysis:** Tomasz Wielek.

**Funding acquisition:** Manuel Schabus.

**Investigation:** Renata Del Giudice, Adelheid Lang.

**Methodology:** Tomasz Wielek, Manuel Schabus.

**Project administration:** Manuel Schabus.

**Resources:** Manuel Schabus.

**Supervision:** Manuel Schabus.

**Visualization:** Tomasz Wielek.

**Writing – original draft:** Tomasz Wielek.

**Writing – review & editing:** Renata Del Giudice, Adelheid Lang, Malgorzata Wislowska, Manuel Schabus.

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
