## [Decision Letter · Decision Letter 0]

16 Jul 2019

PONE-D-19-18265

On the development of sleep states in the first weeks of life

PLOS ONE

Dear Dr. Schabus,

Thank you for submitting your manuscript to PLOS ONE. Your paper was overall well received and definitely has merit. Still, some important issues need to be addressed. Also please note that one reviewer had already submitted their review before receiving your data. As I wrote you per email, please make an extra effort in annotating the data, and in providing instructions for the complete reproducibility of your results. We would appreciate receiving your revised manuscript by Aug 30 2019 11:59PM. To enhance the reproducibility of your results, we recommend that if applicable you deposit your laboratory protocols in protocols.io, where a protocol can be assigned its own identifier (DOI) such that it can be cited independently in the future. For instructions see: http://journals.plos.org/plosone/s/submission-guidelines#loc-laboratory-protocols

We look forward to receiving your revised manuscript.

Kind regards,

Daniele Marinazzo

Academic Editor

PLOS ONE

Journal Requirements:

1. Please amend the subsection category “[FOR JOURNAL STAFF USE ONLY]” for your manuscript. Unfortunately, this is not a valid category. At this time, please choose one or more subsections that best represent the topic(s) of your study.

Reviewers' comments:

Reviewer's Responses to Questions

**Comments to the Author**

1. Is the manuscript technically sound, and do the data support the conclusions?

Reviewer #1: Yes

Reviewer #2: Yes

2. Has the statistical analysis been performed appropriately and rigorously? 

Reviewer #1: I Don't Know

Reviewer #2: Yes

3. Have the authors made all data underlying the findings in their manuscript fully available?

Reviewer #1: No

Reviewer #2: Yes

4. Is the manuscript presented in an intelligible fashion and written in standard English?

Reviewer #1: Yes

Reviewer #2: Yes

5. Review Comments to the Author

Reviewer #1: Overall the paper is well organised, clear, straight to the point and easy to understand.

I only have few minor comments that the authors may consider to keep into account for a revised version of the ms.

I would avoid the term "well-sized sample" when defining the number of newborns included in the study, at least if the authors did not perform any analysis to define the sample size.

The authors report that the MSPE was calculated for non-overlapping 30s segments. Do the authors investigated the possible time-window effect? How much the reported results may depend on this "arbitrary" choice?

The authors state that "To provide equal number of observation for each subject and for each sleep stage MSPE (or PSD) values were averaged first across EEG channels, and next across a random sample of 10 epochs". If I understand correctly, the authors started with a 128 channels EEG and ended with features extracted from the "average" of 6 EEG channels. Isn't this a bit odd? May the authors comment on this?

I would avoid to define "marginally significant" an interaction with p=.086. At least, if they do not consider "marginally no significant" the p-values equal to 0.02...

Probably, the - only - residual main concern (but fully disclosed by the author in the limitations section) is related with the lacking of a double (manual) scoring.

Reviewer #2: Wielek et al. (1) describe the power and multiscale permutation entropy features of babies’ wake, REM and non-REM states as a function of early development and (2) automatically classify those stages with such features. They validate the automatic classification using manual labeling techniques. Characterizing developmental changes in the spectral and complex signatures that differentiate various arousal/sleep states is highly relevant both to developmental science as well as cognitive neuroscience, as is knowledge regarding the practical utility of automatically predicting sleep stages on the basis of these features. The use of spectral and entropy features is well motivated and the associated references are well chosen. Wielek et al. find that across all channels, fast permutation entropy increased from REM to REM and wake, whereas the opposite pattern was observed at slow scales. Furthermore, entropy during sleep decreased with early development, whereas no power or entropy changes were observed during wake. Changes in entropy were not equally observed in different power bins and entropy (mostly from frontal and physiological channels) better recovered the manual sleep stage labels than spectral power. NREM and wake states could be recovered with little confusion, aligning with the observed entropy gradient across arousal states.

Most of the manuscript is clearly framed and motivated, while results are presented in a clear fashion. Nonetheless, I found some of the results difficult to interpret due to methodological and conceptual limitations, which I will describe in more detail below. Given the relevance of the topic, I would recommend the acceptance of this manuscript conditional on changes made referring to the points below.

Major points

1. One general limitation of the approach by Wielek et al. is that it relies on an accurate manual scoring, which the authors sufficiently note in the text. Due to the centrality of accurate scoring, I was surprised that the paper did not describe changes in sleep stages between sessions in more detail and discuss whether and how their results deviate from previous work looking at early development and sleep. Under the assumption that manual sleep scoring is the gold standard, manually labeled sleep stages should already offer insight into development, although I could surprisingly not find results pertaining to e.g. the relative duration of classes.

2. The dual aims of this manuscript (1: describing entropy & PSD changes with arousal states and age; 2: decoding such states) are not perfectly intersected. For example, decoding of sleep stages appears to be maximal from frontal channels, but the description of entropy & PSD features is done on the average across all channels. Therefore, both parts of the manuscript appear rather segmented.

Scoring is described as difficult due to artifacts. This would also affect the features used for manual labeling. Sleep stage scoring appears to be driven primarily by physiological as well as frontal channels that are highly prone to muscular contributions, suggesting that automatic sleep stage scoring may be driven primarily by philological channels as well as frontal channels that are highly prone to muscular contribution. Thus, decoding accuracy may derive to a large part also from non-neural sources, which fundamentally constrains arguments related to ‘cortical development’ This should be more clearly stated and discussed in the manuscript as it challenges the interpretation as stemming from neural sources.

3. More generally, the mechanistic interpretation of the observed entropy effects is difficult here, due to the potential nonlinear contributions to the measure. In contrast to the observed differences in entropy and limited modulation of rhythmic features, much of the discussion focusses on rhythmic signatures. This may relate at least in part to the way these binned values are calculated. A global power normalization is uncommon and produces effects that are very difficult to interpret. Due to the 1/f distribution of the power spectrum, even high frequency power will always be normalized to the predominant low-frequency power. This is readily observed also from the plots (see Figure 4, 1-3Hz has relative power around 90%, whereas beta power is in the range of 5%.) This is especially problematic, as the normalization is applied regardless of sleep stage, which presumably consist of different rations of low-to-high-frequency content. Unequal distributions of sleep stages as observed in the present data may therefore create unequal baselines. It’s further not clear how this normalization would ‘facilitate comparison between session’ (l. 196 f.) as session differences could be expressed in different baselines. Concerns regarding normalization choices could be alleviated by presenting average PSD spectra for the individual sleep stages as has been done in Figure S6 for entropy.

4. Linked to the relevance of considering the entire spectrum rather than narrow bands/bins, preliminary evidence suggests that sleep stages may be differentiated by the slope of the arrhythmic 1/f spectrum (Lendner et al., 2019, bioRxiv). Notably, fast scale entropy is often directly related to 1/f slopes (Bruce et al., 2009, Waschke et al., 2017, Vakorin & McIntosh, 2012), suggesting that a similar link may also exist in the present data. [On a side note, this link of a single scale to a multiscale property such as the 1/f spectrum questions to some extent the notion that prior approaches used ‘temporally more unspecific entropy measures’ (l. 443).] High convergence between these measures would provide more information about the interpretation of fine-scale permutation entropy here.

5. A big advantage of this dataset appears to be the longitudinal (vs. cross-sectional) nature of the design, which was surprisingly not overtly noted. On the negative side, there does not appear to be a habituation session. The session effect could thus at least in part also reflect a retest effect due to habituation effects, which should be noted.

Minor points

-L. 30: ‘the baby’s brain signal complexity (and spectral power) revealed huge developmental changes in sleep in the first 5 weeks of life’. As no effect size measures were provided and effects visually are rather constrained, I would refrain from using the word ‘huge’ here. The same goes for statements such as ‘massive drop’ (l. 458).

-L. 100ff.: ‘a big practical advantage is that entropy-based features, such as permutation

-entropy, are typically more robust against common EEG artifacts as compared to spectral measures (Bandt et al., 2002)]’. This statement and reference are questionable. The reference merely shows that nonlinear features can be robustly identified even in the presence of noise. But no comparison to spectral features is provided. Furthermore, strong noise would also impact permutation entropy, especially if it is strong enough to limit manual labeling as suggested by the authors.

-The section on Participants and EEG should add a description of the criteria for starting and stopping the approx. 30 min recordings were.

-L151f.: “Segments with artifacts were rejected based on simple power spectral density (PSD) thresholds”

oNo information is available regarding what these thresholds were.

-L93: “such as Fast Fourier”. The full title of ‘Fast Fourier Transform’ would be necessary here.

-I found particular aspects of the statistical procedure questionable. First, all data are averaged across EEG channels and then 10 random samples were selected to equate epoch amounts. Here, some sort of random re-sampling (e.g. bootstrapping) should be considered. Regarding the averaging procedure, Figure S4 provides an interesting contrast of frontal and occipital channels. Such a contrast makes sense given that the decoding analysis suggests a stronger representation of sleep stage information at frontal channels. However, no statistics appear to be used, especially no statistics support the claim that the ‘difference between REM and WAKE at week-5 is more pronounced at the frontal channels’ (lines 690ff.). In Figure S1, inference on which stage decoding is improved between features appears to lack statistics.

-Regarding the power results, effects are observed in the beta and delta band. These bands are infamous for muscle artifact contamination in EEG recordings and some strong outliers are present in the data. The possible influence of artifacts should at least be discussed, although it would be even better to supplement such discussion with power spectra of the data.

-The presentation of individual data points is appreciated. However, this reveals that some conditions include clear outliers (e.g. Slow entropy NREM; especially relevant as much of the discussion focusses on the potential relevance of delta oscillations). These should be controlled for in the statistical analysis.

-Figure 1D is supposed to schematically display that permutation entropy at different levels of coarse-graining can differentiate between different sleep stages. It is unclear how these three exemplary traces were chosen, what the error associated with them is etc. Even for a schematic example, this is misleading for inference purposes. Why not plot MSPE values in addition to power values as depicted in plot C? Or accumulate across all time points within sleep stages to get estimates with an indication of the associated error.

-The bars in the upper plots of Figure 5 are missing labels.

-The provided Figures were of a noticeably low resolution. High-quality vector images would be more appealing for final publication.

-‘This confirms that not all changes in EEG complexity are reflected by changes in power’ (l. 449 ff.). This is a very strong statement that cannot be backed up by the data. Differences in the power spectrum (e.g., 1/f slopes) other than the bins tested could also covary with age and sleep stage.

-‘In our case however, this difference in signal-to-noise ratio has been counterbalanced by the sleep specific entropy decrease from week-2 to week-5, which manifests itself as decreased performance in classifying NREM (week-5 training, week-2 test).’ (l. 520 ff.). It is unclear to my why these two observations are claimed to be related? Fast scale entropy appears to decrease during sleep with age both for NREM and REM sleep. Why would this exclusively affect the classification of NREM?

-‘This confirms that developmental changes’ (l. 524). Very strong, and in my view unsupported strength of the conclusion. Better go with ‘suggest’ etc.

-‘whereas stage wake may be (oscillatory-wise) already widely developed’ (l. 525 f.). To the extent that the power measures here can be interpreted as ‘oscillatory’, the results suggest that also the sleep stages are already widely developed as no pairwise session effects were observed for any frequency bin in any sleep stage.

-The current manuscript still contains grammatical errors. Please consider carefully assessing the manuscript again and correcting those in a revision.

omight render rather low reproducibility (89); cf. yield

oduring quite sleep (116); cf. quiet

oThe classifier is later called as MSPE-based (189).

owe restrict our (260); cf. restricted

oat fast scale (311); cf. a fast scale/ fast scales

o…

6. PLOS authors have the option to publish the peer review history of their article (what does this mean?). If published, this will include your full peer review and any attached files.

Reviewer #1: No

Reviewer #2: No

---

## [Author Response · Author response to Decision Letter 0]

10 Sep 2019

Reviewer #1

• I would avoid the term "well-sized sample" when defining the number of newborns included in the study, at least if the authors did not perform any analysis to define the sample size.

Reply: We agree, corrected as suggested. LINES 125-127

• The authors report that the MSPE was calculated for non-overlapping 30s segments. Do the authors investigate the possible time-window effect? How much the reported results may depend on this "arbitrary" choice?

Reply: We computed MSPE for 30s segments (3750 data points) to match the analysis with standard sleep scoring procedure. We did not check the effect of window length empirically. Based on the literature however, we believe that the window-length is sufficient for the analysis. According to the original paper by Bandt on the permutation entropy, window length ‘should be considerably larger’ than the number of possible ordinal patterns [1]. Similarly Cao and colleagues found that windows of 512, 1024 and 2048 data points (EEG segments) give very similar results, and concluded that within this range, the exact choice of the window length is not critical [2]. In our analysis (multiscale permutation entropy) the effective window size depends on the coarse-graining applied (scale) such that the length of the original signal (3750 data points) gets reduced by the scale factor. The number of possible patterns is always 6. Thus for all degrees of coarse-graining (scales) applied, the above requirements are fulfilled. In this exemplary calculation we calculate window length (W) for the largest time scale used (scale=5):

- scale=5

- W = 3750/5 

- W >> 6 and 512 < W < 2048

Hypothetically considered, if the window length were too short, the computed signal entropy would be underestimated (due to inaccurate estimation of the distribution of patterns). In our analysis such a bias would affect especially slow time scale MSPE (scale=5) rather than fast time scale MSPE (scale=1) as the window length is more strongly reduced in the former case (3750/5 compared to 3750/1).

1. Bandt C, Pompe B. Permutation entropy: a natural complexity measure for time series. Phys Rev Lett. 2002;88(17):174102.2.

2. Cao Y, Tung WW, Gao JB, Protopopescu VA, Hively LM. Detecting dynamical changes in time series using the permutation entropy. Phys Rev E Stat Nonlin Soft Matter Phys. 2004;70(4 Pt 2):046217.

• The authors state that "To provide equal number of observations for each subject and for each sleep stage MSPE (or PSD) values were averaged first across EEG channels, and next across a random sample of 10 epochs". If I understand correctly, the authors started with a 128 channels EEG and ended with features extracted from the "average" of 6 EEG channels. Isn't this a bit odd? May the authors comment on this?

Reply: 

We thank the reviewer for this comment. The reduction in the number of channels (128 to 6 channels) was actually performed in order to mimic the usual recordings available at this age and in clinical settings in general. In addition we want to compare our automated sleep staging to manual sleep scoring which always relies on a set of 2-6 electrodes on the brain level (C3 + C4 in the old Rechtschaffen and Kales standard, 2 frontal, 2 central and 2 occipital in the newer AASM standard). We agree with the reviewer that running statistics on the averages of 6 EEG channels and using 11 channels for the classification was inconsistent. In the manuscript we now do not anymore average across channels but added the channel locations as an additional independent variable to the statistical model (LINES 234-235). Also we updated Table 1 and Fig. 3 in order to illustrate the results for frontal channels (F3 and F4); central and occipital sites can be seen in the supplementary material (S4 Fig. and S5 Fig.). 

Topo-plots based on the full 128 channels setup are now presented in addition in the supplements and show that the selected 6 channels sufficiently capture the spatial patterns (S9 Fig.). However given the exact cap placement and movement of babies at this age we would warn from over-interpreting these results.

• I would avoid to define "marginally significant" an interaction with p=.086. At least, if they do not consider "marginally no significant" the p-values equal to 0.02...

Reply: 

We corrected as suggested. LINE 333

Reviewer #2

MAJOR POINTS

• One general limitation of the approach by Wielek et al. is that it relies on an accurate manual scoring, which the authors sufficiently note in the text. Due to the centrality of accurate scoring, I was surprised that the paper did not describe changes in sleep stages between sessions in more detail and discuss whether and how their results deviate from previous work looking at early development and sleep. Under the assumption that manual sleep scoring is the gold standard, manually labeled sleep stages should already offer insight into development, although I could surprisingly not find results pertaining to e.g. the relative duration of classes.

Reply: 

We fully agree with the reviewer that this part was not sufficiently adressed in our earlier version. We now have added further details (LINES 306-313). 

The revised manuscript reads:

On average 5-week old newborns spend a higher percentage of total time (19%) in NREM sleep as compared with 2-weeks old babies (11.5%). In contrast relative REM duration decreases from 60.6% at week-2 to 57.2%, at week-5, and WAKE decreases from 27.9% at week-2 to 23.8 at week-5. Statistically a significantly larger proportion of participants showed NREM during week-5 (66%) as compared to week-2 (35%) (χ² (1) = 6.81, p < .05). Using paired-samples (by including only those subjects that actually show given sleep-states in both session), we found no significant differences in the median duration of classes from week-2 to week-5

The corresponding discussion part has been altered on lines 500-507.

The revised discussion part reads:

Generally newborns sleep up to 16-17 hours a day. We found that the most dominant sleep stage is REM (week-2: 60.6%, week-5: 57.2% of total time). Indeed, several studies show that newborns spend more than half their time in REM [2]. Its proportion, however, gradually decreases within the first 12 weeks of life [38], which is interpreted as an ongoing adaptation of the sensory system to the environment [2]. In contrast the mean percentage of time for NREM increases from week-2 (11.5%) to week-5 (19%). This proportional increase in NREM reflects a gradual shift towards the adult pattern known as slow wave sleep (SWS) predominance [2]. 

• The dual aims of this manuscript (1: describing entropy & PSD changes with arousal states and age; 2: decoding such states) are not perfectly intersected. For example, decoding of sleep stages appears to be maximal from frontal channels, but the description of entropy & PSD features is done on the average across all channels. Therefore, both parts of the manuscript appear rather segmented.

Reply: We thank the reviewer for this remark and agree that the two sections were a bit inconsistent. Now, instead of averaging across EEG channels, we add brain location (frontal, central, occipital) as an additional independent variable into the statistical model (LINES 234-235). Also, Table 1 and Fig. 3 have been modified in order to illustrate the results for frontal channels (F3 and F4), whereas central and occipital sites have been included in the supplements (S4 Fig. and S5 Fig.).

• Scoring is described as difficult due to artifacts. This would also affect the features used for manual labeling. Sleep stage scoring appears to be driven primarily by physiological as well as frontal channels that are highly prone to muscular contributions, suggesting that automatic sleep stage scoring may be driven primarily by philological channels as well as frontal channels that are highly prone to muscular contribution. Thus, decoding accuracy may derive to a large part also from non-neural sources, which fundamentally constrains arguments related to ‘cortical development’ This should be more clearly stated and discussed in the manuscript as it challenges the interpretation as stemming from neural sources.

Reply: In general we agree with the reviewer. We have provided further discussion on possible confounding factors. LINES 504-509. But please also note that epochs with major artifacts were not scored by the manual scorer (not classified) and consequently not included in the analysis.

The revised manuscript reads:

The fact that frontal channels have been identified as the ones with lowest entropy (fast temporal scale), could be related to infant (1.5 - 6 months) sleep spindles which are prominent over the fronto-central area [44]. Nevertheless, this result needs to be interpreted with caution. It is widely acknowledged that frontal EEG channels are impacted most by eye blinks and muscles artifacts. Despite using robust entropy measure, we cannot exclude the possibility that some part of the effect may be driven by non-neural sources.

• More generally, the mechanistic interpretation of the observed entropy effects is difficult here, due to the potential nonlinear contributions to the measure. In contrast to the observed differences in entropy and limited modulation of rhythmic features, much of the discussion focusses on rhythmic signatures. This may relate at least in part to the way these binned values are calculated. A global power normalization is uncommon and produces effects that are very difficult to interpret. Due to the 1/f distribution of the power spectrum, even high frequency power will always be normalized to the predominant low-frequency power. This is readily observed also from the plots (see Figure 4, 1-3Hz has relative power around 90%, whereas beta power is in the range of 5%.) This is especially problematic, as the normalization is applied regardless of sleep stage, which presumably consists of different ratios of low-to-high-frequency content. Unequal distributions of sleep stages as observed in the present data may therefore create unequal baselines. It’s further not clear how this normalization would ‘facilitate comparison between session’ (l. 196 f.) as session differences could be expressed in different baselines. Concerns regarding normalization choices could be alleviated by presenting average PSD spectra for the individual sleep stages as has been done in Figure S6 for entropy.

Reply: We thank the reviewer for the thoughtful comment. As suggested, we present average PSD spectra instead of the normalized frequency bins. LINES 207-213. 

Admittedly, the implemented corrections made the interpretation easier. We modified the discussion session accordingly. LINES 498-506.

The revised manuscript reads:

This change was observed not only at the fast temporal scale, (mixture of low and high frequencies), but also at the slow temporal scale (slow frequencies only), suggesting a specific bandwidth or temporal scale being involved. Indeed, we also observed a significant increase in EEG spectral power at 1-15Hz (during NREM). The observed entropy decrease and corresponding power increase is likely linked to the emergence of sleep spindles taking place between week-2 and week-8 after birth [3]. The fact that frontal channels have been identified as the ones with lowest entropy (fast temporal scale) could be related to infant (1.5 - 6 months) sleep spindles which are prominent over the fronto-central area [4]

• Linked to the relevance of considering the entire spectrum rather than narrow bands/bins, preliminary evidence suggests that sleep stages may be differentiated by the slope of the arrhythmic 1/f spectrum (Lendner et al., 2019, bioRxiv). Notably, fast scale entropy is often directly related to 1/f slopes (Bruce et al., 2009, Waschke et al., 2017, Vakorin & McIntosh, 2012), suggesting that a similar link may also exist in the present data. [On a side note, this link of a single scale to a multiscale property such as the 1/f spectrum questions to some extent the notion that prior approaches used ‘temporally more unspecific entropy measures’ (l. 443).] High convergence between these measures would provide more information about the interpretation of fine-scale permutation entropy here.

Reply: We thank the reviewer for this comment and added a sentence to the discussion (LINES 541-543):

In relation to PSD data, we observed that high frequency spectral components correspond to higher entropy (when compared across sleep stages), which agrees with previous studies suggesting a link between fast scale entropy and PSD slope [20, 25]

• 5. A big advantage of this dataset appears to be the longitudinal (vs. cross-sectional) nature of the design, which was surprisingly not overtly noted. On the negative side, there does not appear to be a habituation session. The session effect could thus at least in part also reflect a retest effect due to habituation effects, which should be noted.

Reply: We include additional information on data sample in the Participants and methods section (LINES 150-153):

Recording times were determined by the experimental protocol including nine 3min or 5min periods of alternating rest and auditory stimulation periods (with simple nursery rhymes). For the current study we disregard this experimental stimulation and merely focus on the changes in behavioral states over the full recording time.

Second, we put an additional clarification at the beginning of the discussion (LINES 471-477). The text reads:

Please note that the main focus of this study was the development of an automatic sleep staging technique in order to sleep stage newborn sleep as early as 2 to 5 weeks after birth. As babies at this age also sleep a significant proportion of time (and irrespective of environmental stimulation such as noise) we also had the opportunity to study sleep and associated brain dynamics in this early age group. However, please note that we do not claim that our newborn sleep data is necessarily representing natural sleep at that early age as auditory stimulation is ongoing half of the time in our study protocol.

MINOR POINTS

• -‘the baby’s brain signal complexity (and spectral power) revealed huge developmental changes in sleep in the first 5 weeks of life’. As no effect size measures were provided and effects visually are rather constrained, I would refrain from using the word ‘huge’ here. The same goes for statements such as ‘massive drop’ (l. 458).

Reply: We corrected as suggested. LINE 30 and LINE 523

• ‘a big practical advantage is that entropy-based features, such as permutation

-entropy, are typically more robust against common EEG artifacts as compared to spectral measures (Bandt et al., 2002)]’. This statement and reference are questionable. The reference merely shows that nonlinear features can be robustly identified even in the presence of noise. But no comparison to spectral features is provided. Furthermore, strong noise would also impact permutation entropy, especially if it is strong enough to limit manual labeling as suggested by the authors.

Reply: We agree with the reviewer’s assessment that a direct comparison is not possible here. We corrected the reference and rephrased the sentence by emphasizing the robustness due to symbolic/ordinal nature of permutation entropy. LINES 100-104. 

It read as follows:

In contrast to FFT-based measures, symbolic measures such as permutation entropy are operating on the order of values rather than on the absolute values of a time series. This has a big practical advantage if a signal is highly non-stationary and corrupted by noise [5], as is the case with the data of newborns. For instance, noise due to high electrode impedance is less likely to affect symbolic measures such as permutation entropy [6]. 

• The section on Participants and EEG should add a description of the criteria for starting and stopping the approx. 30 min recordings were.

Reply: We corrected as suggested. LINES 149-153. The revised manuscript reads:

The signal was recorded continuously with a sampling rate of 500Hz over 35min (n=11) or 27min (n=31). Recording times were determined by the experimental protocol including nine 3min or 5min periods of alternating rest and auditory stimulation periods with simple nursery rhymes. For the current study we disregard this experimental stimulation and merely focus on the changes in behavioral states over the full recording time.

• “Segments with artifacts were rejected based on simple power spectral density (PSD) thresholds”

No information is available regarding what these thresholds were.

Reply: We added information as suggested; L160-164. In the manuscript it read as follows:

Electrode (impedance check) artifacts characterized by a 20Hz component were deleted semi-automatically by first visually inspecting individual recordings in the time-frequency domain and next iterating over segments. Percentile thresholding was used to exclude bad segments which resulted in an exclusion of 4.5% of total segments.

• “such as Fast Fourier”. The full title of ‘Fast Fourier Transform’ would be necessary here.

Reply: Corrected as suggested. LINE 93.

• I found particular aspects of the statistical procedure questionable. First, all data are averaged across EEG channels and then 10 random samples were selected to equate epoch amounts. Here, some sort of random re-sampling (e.g. bootstrapping) should be considered. Regarding the averaging procedure, Figure S4 provides an interesting contrast of frontal and occipital channels. Such a contrast makes sense given that the decoding analysis suggests a stronger representation of sleep stage information at frontal channels. However, no statistics appear to be used, especially no statistics support the claim that the ‘difference between REM and WAKE at week-5 is more pronounced at the frontal channels’ (lines 690ff.). In Figure S1, inference on which stage decoding is improved between features appears to lack statistics.

Reply: We thank the reviewer again for the very constructive comments. We implemented random re-sampling as suggested. Its implementation (python function named mybootstraper) can be found in the two scripts: 4_run_plot_mspe.py and 7_run_plot_psd.py. Both statistics and figure have been re-computed using the new sampling scheme. With regards to channels, we have included locations as an additional factor in the statistical model (please also see our reply to the major point 2). The sentence ‘difference between REM and WAKE at week-5 is more pronounced at the frontal channels’, has been deleted as the 3-way interaction was not significant. Regarding Figure S1, we now provide additional statistics as suggested. 

The revised manuscript reads: 

LINES 232-235: To provide an equal number of observations for each subject and for each sleep stage a bootstrap sample of 10 MSPE values was repeatedly (1000 times) drawn and eventually averaged. 

LINES 774-776: Note that overall entropy at the fast temporal scale is lower over frontal (see Fig. 3, main text) as compared to both central and occipital channels (left panels).

• Regarding the power results, effects are observed in the beta and delta band. These bands are infamous for muscle artifact contamination in EEG recordings and some strong outliers are present in the data. The possible influence of artifacts should at least be discussed, although it would be even better to supplement such discussion with power spectra of the data.

Reply: We thank the reviewer for the comment and as suggested included average power spectra (Fig. 4 for frontal and S5 Fig. for both central and occipital channels). Also, as described above, we discuss possible influence of frontal artifacts.

• The presentation of individual data points is appreciated. However, this reveals that some conditions include clear outliers (e.g. Slow entropy NREM; especially relevant as much of the discussion focusses on the potential relevance of delta oscillations). These should be controlled for in the statistical analysis.

Reply: We agree with the reviewer. We drop outliers (as by the interquartile range rule), repeat the statistical analysis and if inferences (with and without outliers) differ, report both results. 

The revised manuscript reads:

LINES 242-245: Additionally, we carefully report any differences if the exclusion of statistical outliers (as identified by the interquartile range rule) changed results significantly.

LINES 349-352: Note that session x sleep stage interaction changed to a trend when excluding outliers (based on the interquartile rule) (Wald chi-square (2, 73) = 4.67, p=.09).

• Figure 1D is supposed to schematically display that permutation entropy at different levels of coarse-graining can differentiate between different sleep stages. It is unclear how these three exemplary traces were chosen, what the error associated with them is etc. Even for a schematic example, this is misleading for inference purposes. Why not plot MSPE values in addition to power values as depicted in plot C? Or accumulate across all time points within sleep stages to get estimates with an indication of the associated error.

Reply: We thank the reviewer for this suggestion. As proposed, instead of presenting single epochs, we accumulate across time. Also color-coding of sleep classes was used for a better visibility. 

• The bars in the upper plots of Figure 5 are missing labels.

Reply: We corrected as suggested, the revised manuscript reads (LINE 443): 

95% confidence-intervals (for both accuracy and F1 scores) are displayed on the basis of bootstrap analysis.

• The provided Figures were of a noticeably low resolution. High-quality vector images would be more appealing for final publication.

Reply: We corrected as suggested

• ‘This confirms that not all changes in EEG complexity are reflected by changes in power’ (l. 449 ff.). This is a very strong statement that cannot be backed up by the data. Differences in the power spectrum (e.g., 1/f slopes) other than the bins tested could also covary with age and sleep stage.

Reply: We agree. The discussion section has been reorganized accordingly, LINES 495-509. The aforementioned statement was deleted. 

• ‘In our case however, this difference in signal-to-noise ratio has been counterbalanced by the sleep specific entropy decrease from week-2 to week-5, which manifests itself as decreased performance in classifying NREM (week-5 training, week-2 test).’ (l. 520 ff.). It is unclear to me why these two observations are claimed to be related? Fast scale entropy appears to decrease during sleep with age both for NREM and REM sleep. Why would this exclusively affect the classification of NREM?

Reply: We agree that the statement was imprecise as accuracy for REM decreases as well. The sentence has been changed and now reads (LINES 588-592):

In our case however, this difference in signal-to-noise ratio has been counterbalanced by the sleep specific entropy decrease from week-2 to week-5, which manifests itself as decreased performance in classifying both NREM (week-5 training, week-2 test) and REM (week-2 training, week5-test) with a 24% and 14% accuracy drop, respectively.

• ‘This confirms that developmental changes’ (l. 524). Very strong, and in my view unsupported strength of the conclusion. Better go with ‘suggest’ etc.

Reply: Corrected as suggested. LINE 593

• ‘whereas stage wake may be (oscillatory-wise) already widely developed’ (l. 525 f.). To the extent that the power measures here can be interpreted as ‘oscillatory’, the results suggest that also the sleep stages are already widely developed as no pairwise session effects were observed for any frequency bin in any sleep stage.

Reply: We agree. As the sentence refers to ‘signal complexity’ mentioning oscillations was misleading. Corrected as suggested. Please note, however, that after the suggested change in PSD analysis, we observe session effect for NREM. LINE 594

• The current manuscript still contains grammatical errors. Please consider carefully assessing the manuscript again and correcting those in a revision.

omight render rather low reproducibility (89); cf. yield

Reply: Corrected as suggested. LINE 89

• oduring quite sleep (116); cf. quiet

Reply: Corrected as suggested. LINE 113 and LINE 121

• The classifier is latr called as MSPE-based (189).

owe restrict our (260); cf. restricted

Reply: Corrected as suggested. LINE 280

• fast scale (311); cf. a fast scale/ fast scales

Reply: Corrected as suggested. LINE 351

---

## [Decision Letter · Decision Letter 1]

25 Sep 2019

PONE-D-19-18265R1

On the development of sleep states in the first weeks of life

PLOS ONE

Dear Dr. Schabus,

Thank you for submitting your revised manuscript to PLOS ONE.

Your way of addressing the issues emerging upon the first submission was overall very appreciated. A few points, detailed in the reviews below, remain to be clarified before we can accept this paper in PLOS One.

We would appreciate receiving your revised manuscript by Nov 09 2019 11:59PM. To enhance the reproducibility of your results, we recommend that if applicable you deposit your laboratory protocols in protocols.io, where a protocol can be assigned its own identifier (DOI) such that it can be cited independently in the future. For instructions see: http://journals.plos.org/plosone/s/submission-guidelines#loc-laboratory-protocols

We look forward to receiving your revised manuscript.

Kind regards,

Daniele Marinazzo

Academic Editor

PLOS ONE

Reviewers' comments:

Reviewer's Responses to Questions

**Comments to the Author**

1. If the authors have adequately addressed your comments raised in a previous round of review and you feel that this manuscript is now acceptable for publication, you may indicate that here to bypass the “Comments to the Author” section, enter your conflict of interest statement in the “Confidential to Editor” section, and submit your "Accept" recommendation.

Reviewer #1: All comments have been addressed

Reviewer #2: (No Response)

2. Is the manuscript technically sound, and do the data support the conclusions?

Reviewer #1: Yes

Reviewer #2: Partly

3. Has the statistical analysis been performed appropriately and rigorously? 

Reviewer #1: Yes

Reviewer #2: Yes

4. Have the authors made all data underlying the findings in their manuscript fully available?

Reviewer #1: Yes

Reviewer #2: Yes

5. Is the manuscript presented in an intelligible fashion and written in standard English?

Reviewer #1: Yes

Reviewer #2: Yes

6. Review Comments to the Author

Reviewer #1: I have not further comments. All my previous comments have been addressed. The manuscript is technically sound and clearly presented.

Reviewer #2: Many thanks to the authors for the revision of the manuscript. The changes addressed many of my previous concerns and the current manuscript presents the data and their limitations more clearly. In addition, the available data documentation appears clear, although I did not run any of the scripts. There are a few points that I would still like to see addressed, including some changes introduces in the current revision. With such minor revisions, I would recommend publication in PLOS ONE.

- A more thorough discussion of decoding sleep stages from physiological channels is still missing. While the potential influence of physiological noise on the interpretation of frontal EEG recordings has been adequately noted in multiple places, the decoding results warrant a specific discussion. I recommend noting that development of sleep patterns is not only associated with neural, but also with physiological changes (e.g., eye movement output) and that these features may be crucial markers for the manually labeled sleep stages. With the provided data, it appears wrong for readers to infer that neural differences are driving classification results and this should be made explicit.

- “On average 5-weeks old newborns spend a higher percentage of total time (19%) in NREM sleep as compared with 2-weeks old babies (11.5%). […] A significantly larger proportion of participants showed NREM during week-5 (66%) as compared to week-2 (35%) […] Using paired-samples (by including only those subjects that actually show given sleep-state in both sessions), we found no significant effects differences in the median duration of classes from week-2 to week-5”. Were pair-wise statistics performed on the initial (overall) data? If these differences are significant, wouldn’t the inference be that cross-sectionally, NREM sleep increases, but not longitudinally within person? If so, this should be noted as much of the discussion ascribes the observed neural effects to an increase in NREM sleep and an associated increase in slow waves.

- Thank you for adding information on when the recordings were taken. I am still a bit unclear about the exact recording setup. Were mothers asked to perform nursery rhymes as acoustic stimulation every approx. 5 min? More information on this would be appreciated. The fact that the data were not recorded during continuous night-time sleep should also be repeated in the Introduction and Discussion section to avoid misunderstandings.

- Figure S9 is unclear to me. Why are the PSD curves repeated for each bin when they are identical? I am also not sure what the Figure is telling me about the presented bins as no statistical comparison is made here. This is complicated by the fact that value ranges are largely unreadable.

- The discussion of sleep spindle development and anticorrelated entropy is much clearer now. Given this discussion, it would be interesting to know whether spectral differences between sessions were inter-individually anticorrelated with differences in entropy, i.e., whether spectral features relate to observed entropy effects.

- 163: ‘Percentile thresholding’. 95% percentile?

- 224: ‘individual amount’. I’d recommend ‘inter-individual differences.

- 286: ‘paired-samples’. Specific that those are ‘paired-samples t-tests’.

- 347: “Note that only NREM shows differences in the PSD spectra between age groups and the developing 9–14 Hz peak exclusively observed at week-5 of age.” Differences in the alpha peak are hard to see and are more obvious at occipital channels (Figure S5). A log-log scaling in addition with error bars may aid visualization.

- 359: “reveled” vs. ‘revealed’

- 376: “We found an increase in oscillatory activity from week-2 to week-5”. I would recommend to change this to spectral power, as this does not appear to be a narrowband, i.e. oscillatory, effect. Also, the addition of error bars to Figure 4 would be beneficial.

- 471: Starting the discussion with “Please note that” appears unnecessarily defensive and should be dropped.

- 745: “Given the rapid cap placement and movement of babies at this early age we would refrain from over-interpreting these results.” This is a puzzling statement to me, at least in its current phrasing. Are the authors concerned that cap positioning strongly varied across time points and babies? Then all of the other results in the manuscript would be equally questionable as the inferences assume that the selected channels are in comparable locations.

7. PLOS authors have the option to publish the peer review history of their article (what does this mean?). If published, this will include your full peer review and any attached files.

Reviewer #1: No

Reviewer #2: No

---

## [Author Response · Author response to Decision Letter 1]

14 Oct 2019

1. A more thorough discussion of decoding sleep stages from physiological channels is still missing. While the potential influence of physiological noise on the interpretation of frontal EEG recordings has been adequately noted in multiple places, the decoding results warrant a specific discussion. I recommend noting that development of sleep patterns is not only associated with neural, but also with physiological changes (e.g., eye movement output) and that these features may be crucial markers for the manually labeled sleep stages. With the provided data, it appears wrong for readers to infer that neural differences are driving classification results and this should be made explicit.

Reply: We agree with the reviewer and thank for the suggestion. The revised manuscript (LINE 555-558) reads:

“Finally yet importantly we found that horizontal EOGs, frontal brain channels as well ECG contribute the most in the classification. This suggests that development of sleep patterns is not only associated with neural, but also with physiological changes (e.g., eye movement) and that these features may be crucial markers for the manually labeled sleep stages.”

2. “On average 5-weeks old newborns spend a higher percentage of total time (19%) in NREM sleep as compared with 2-weeks old babies (11.5%). […] A significantly larger proportion of participants showed NREM during week-5 (66%) as compared to week-2 (35%) […] Using paired-samples (by including only those subjects that actually show given sleep-state in both sessions), we found no significant effects differences in the median duration of classes from week-2 to week-5”. Were pair-wise statistics performed on the initial (overall) data? If these differences are significant, wouldn’t the inference be that cross-sectionally, NREM sleep increases, but not longitudinally within person? If so, this should be noted as much of the discussion ascribes the observed neural effects to an increase in NREM sleep and an associated increase in slow waves.

Reply: To compare time in a given sleep stage across sessions we used ‘standard’ pair-wise statistics which limits sample to only those subjects that actually show given sleep stage in both sessions (and in consequence reduces sensitivity/statistical power). I contrast, testing entropy changes we used more sophisticated mixed models, better preserving statistical power in the presence of missing data. We agree however that additional comment would be useful here and add following sentence in the Discussion. LINE 454-457:

 “However, it is worth mentioning that testing longitudinally within subject, we found no significant changes in the percentage of NREM. It was likely due to limited sensitivity in this measure, magnified by small number of available subjects with NREM during week 2. “

3. Thank you for adding information on when the recordings were taken. I am still a bit unclear about the exact recording setup. Were mothers asked to perform nursery rhymes as acoustic stimulation every approx. 5 min? More information on this would be appreciated. The fact that the data were not recorded during continuous night-time sleep should also be repeated in the Introduction and Discussion section to avoid misunderstandings.

Reply: The presented auditory stimuli were pre-recorded, so mothers were not performing any task during PSD recording of their babies. The 3min or 5min periods mentioned in the text were achieved by repetitive presentation of a nursery rhyme that lasted approx. 41s each. We have modified the method section; LINE 148. Also additional clarification has been added in the Introduction (LINE123-125): 

“It has to be mentioned that the reported data were not recorded during continuous night-time sleep period, thus may differ from natural sleep in newborns.”

With regard to the Discussion we believe that its first paragraph already provides sufficient clarification in this respect, thus no further changes were implemented. It reads as follows:

“The main focus of this study was the development of an automatic sleep staging technique in order to sleep stage newborn sleep as early as 2 to 5 weeks after birth. As babies at this age also sleep a significant proportion of time (and irrespective of environmental stimulation such as noise) we also had the opportunity to study sleep and associated brain dynamics in this early age group. However, please note that we do not claim that our newborn sleep data is necessarily representing natural sleep at that early age as auditory stimulation is ongoing half of the time in our study protocol.” 

4. Figure S9 is unclear to me. Why are the PSD curves repeated for each bin when they are identical? I am also not sure what the Figure is telling me about the presented bins as no statistical comparison is made here. This is complicated by the fact that value ranges are largely unreadable.

Reply: We agree that Figure S9 presenting topo-plots and repeated psd curves is largely redundant. We decided to replace it with a new Figure S9 that shows correlations between sigma power and entropy values (please see the response to the point below). 

5. The discussion of sleep spindle development and anticorrelated entropy is much clearer now. Given this discussion, it would be interesting to know whether spectral differences between sessions were inter-individually anticorrelated with differences in entropy, i.e., whether spectral features relate to observed entropy effects.

Reply: We thank the reviewer for this comment. As suggested we show correlations between spectral features and entropy at fast temporal scale (samples from the 2 sessions were merged). Suggested analysis for relative values (differences between sessions) was inconclusive due to small number of available subjects with NREM during week 2. LINE 761-764

6. ‘Percentile thresholding’. 95% percentile?

Reply: Yes, that's right, we specified. LINE 158

7. ‘individual amount’. I’d recommend ‘inter-individual differences.

Reply: We thank for this comment. We corrected as suggested. LINE 226

8. ‘paired-samples’. Specific that those are ‘paired-samples t-tests’.

Reply: We specified as suggested ‘Using paired-samples Wilcoxon test’. LINE 288

9. “Note that only NREM shows differences in the PSD spectra between age groups and the developing 9–14 Hz peak exclusively observed at week-5 of age.” Differences in the alpha peak are hard to see and are more obvious at occipital channels (Figure S5). A log-log scaling in addition with error bars may aid visualization.

Reply: We thank for this suggestion. We modified both Fig4 and S5_Fig by adding log-log scaling.

10. “reveled” vs. ‘revealed’

Reply: We corrected as suggested. LINE 329

11. “We found an increase in oscillatory activity from week-2 to week-5”. I would recommend to change this to spectral power, as this does not appear to be a narrowband, i.e. oscillatory, effect. Also, the addition of error bars to Figure 4 would be beneficial.

Reply: We corrected as suggested; LINE 343. Also we provided standard error of the mean to Figure 4.

12. Starting the discussion with “Please note that” appears unnecessarily defensive and should be dropped.

Reply: We thank the reviewer for this comment, we corrected as suggested. LINE 417

13. “Given the rapid cap placement and movement of babies at this early age we would refrain from over-interpreting these results.” This is a puzzling statement to me, at least in its current phrasing. Are the authors concerned that cap positioning strongly varied across time points and babies? Then all of the other results in the manuscript would be equally questionable as the inferences assume that the selected channels are in comparable locations.

Reply: We agree that the wording was confusing. There was no variation in cap positioning. For the reasons mentioned above, we decided not to include previous version of Figure S9 in the already lengthy supplementary materials.

---

## [Editor Report · Decision Letter 2]

16 Oct 2019

On the development of sleep states in the first weeks of life

PONE-D-19-18265R2

Dear Dr. Schabus,

We are pleased to inform you that your manuscript has been judged scientifically suitable for publication and will be formally accepted for publication once it complies with all outstanding technical requirements.

With kind regards,

Daniele Marinazzo

Academic Editor

PLOS ONE
---

## [Editor Report · Acceptance letter]

21 Oct 2019

PONE-D-19-18265R2 

On the development of sleep states in the first weeks of life 

Dear Dr. Schabus:

I am pleased to inform you that your manuscript has been deemed suitable for publication in PLOS ONE. Congratulations! Your manuscript is now with our production department. 

With kind regards,

on behalf of

Dr. Daniele Marinazzo 

Academic Editor

PLOS ONE